# Athletes’ Knowledge of Pelvic Floor Dysfunction and Their Knowledge of and Engagement with Pelvic Floor Muscle Training: A Scoping Review

**DOI:** 10.3390/ijerph22010104

**Published:** 2025-01-14

**Authors:** Jacinta Magor, Romany Martin, Marie-Louise Bird

**Affiliations:** Department of Physiotherapy, School of Health Science, College of Health and Medicine, University of Tasmania, Launceston 7250, Australia; bainjm@utas.edu.au (J.M.); romany.martin@utas.edu.au (R.M.)

**Keywords:** athlete, sport, incontinence, pelvic floor dysfunction, pelvic floor muscle training

## Abstract

Introduction: Pelvic floor dysfunction (PFD) is prevalent among athletes. Investigating whether athletes are practicing pelvic floor muscle training (PFMT) will assist in delineating the factors underlying the burden of PFD in this population. Additionally, investigating athletes’ knowledge of PFD and knowledge of and attitudes toward PFMT may inform interventions to improve the practice of PFMT. Aims: This scoping review aimed to collate the available evidence regarding athletes’ knowledge of PFD and their knowledge of and engagement with PFMT. Furthermore, this review aimed to determine the types of athletes (sports and competition levels) research had been conducted on and the definitions of PFD and PFMT most used. Methods: Six databases were searched up to January 2024. No study design or publication types were restricted; however, non-English articles were excluded due to resource constraints. Additional publications were identified through the reference lists of included articles. Data were synthesized and presented under subheadings relevant to the aims of this review. Results: Thirty-five publications were included for data extraction. Athletes had low practice of PFMT, poor knowledge of PFD and PFMT, but positive attitudes toward PFMT. One publication reported that higher levels of knowledge were associated with a lower prevalence of PFD. The athletic population was heterogeneous in sport type and competition level, and the definitions of PFD and PFMT were ill-defined. Conclusions: Athletes have a low practice of PFMT and poor knowledge of PFD and PFMT. The education of athletes may have a role in supporting pelvic floor health.

## 1. Introduction

Pelvic floor dysfunction (PFD) refers to conditions that lead to moderate-to-severe impairment of the pelvic floor (PF) [1]. The symptoms of PFD can include urinary incontinence (UI), pelvic organ prolapse (POP), anal incontinence (AI), lumbopelvic pain, and sexual dysfunction [2]. Males and females can experience PFD, with PFD being more prevalent in athletes than non-athletes [3]. In some athletes, PFD is associated with reduced quality of life and impaired sports performance [4,5].

Pelvic floor muscle training (PFMT) is an exercise designed to improve pelvic floor muscle strength, endurance, power, and relaxation [6]. When practiced, PFMT has been found to decrease PFD and improve quality of life [7]. Some authors suggest that greater benefits could be seen if PFMT was included in athletes’ general strength training programs [8].

Despite the benefits of PFMT, there is limited research on athletes’ practice of PFMT in different sports [1]. Additionally, coaches do not have the skills to advise on pelvic floor muscles within their scope of practice [9]. The dearth of research means the relationship between athletes’ PFMT practice and the PFD they experience is unclear. Athletes may practice PFMT, and their PFD may remain due to other factors [10]. Alternatively, as hypothesized by Stickley et al. (2023), low levels of practice of PFMT may be contributing to athletes’ high prevalence of PFD [11]. If athletes’ practice of PFMT is low, improving their practice may decrease PFD, and subsequently, strategies for increasing their practice of PFMT may become treatment priorities for health professionals. Understanding athletes’ practice of PFMT would assist in delineating the factors associated with their significant burden of PFD and inform the role of health professionals in addressing PFD.

Similarly, the literature is lacking regarding athletes’ knowledge of PFD and PFMT and their attitudes toward PFMT. Knowledge and attitudes may influence the prevalence of PFD through their effect on the practice of PFMT. Without knowledge of PFMT, athletes cannot complete the exercises, and their practice of PFMT may reduce if they have inaccurate knowledge [12]. Additionally, attitudinal change is considered the antecedent to behavioral change in behavior change models [13]. Athletes’ attitudes toward PFMT arise as a topic of interest as attitudinal change may be required to facilitate the practice of PFMT. Jaffar et al. (2022) supported these points when they determined that both knowledge of and attitudes toward PFMT were significantly associated with PFD in pregnant women and should be addressed when managing UI [14]. In a non-athletic population, educational interventions have been suggested to improve women’s knowledge and attitudes to increase adherence to PFMT programs and decrease PFD [12]. Understanding athletes’ knowledge of PFD and PFMT and attitudes toward PFMT may inform appropriate educational interventions that might improve athletes’ practice of PFMT and subsequently reduce the negative impact of PFD.

With acknowledgment of the benefit of PFMT in reducing PFD [7] and the burden of PFD in athletes [3,4], investigating why the high rates of PFD prevail in this population is an area that warrants research. This scoping review aimed to address the above gaps and collate the available evidence regarding athletes’ knowledge of PFD and their knowledge of and engagement with PFMT. Furthermore, this review aimed to ascertain the types of athletes (sports and competition level) described in published research and synthesize the study characteristics, outcomes, and definitions of ‘PFD’ and ‘PFMT’ within the identified literature.

## 2. Materials and Methods

### 2.1. Review Typology

A scoping review was chosen to determine the extent of the literature available on the topic and identify concepts that need to be researched further. This typology enabled the working definitions of ‘PFD’ and ‘PFMT’ to be clarified so that the boundaries of this topic can be understood. The Johanna Briggs Institute scoping manual was used to guide the authors in preparing this review. The PRISMA Scoping Reviews Checklist was used to ensure systematic and thorough reporting [15].

### 2.2. Research Question

The following primary research question was developed: “What is athletes’ knowledge of PFD and PFMT and what is their engagement with PFMT?”.

One sub-question was developed to describe the relevant population:

“What type of athletes (what sports and competition level) has this research been conducted on?”.

### 2.3. Eligibility Criteria

Studies were eligible for inclusion in this review if they met the following population, concept, and context criteria:-Population: athletes as defined by their involvement in competitive sports or as identified as athletes within the study;-Concept: knowledge of PF function or PFD or knowledge of and engagement with PFMT. Engagement with PFMT was defined as concerning athletes’ attitudes toward PFMT and their practice (habitual application) of PFMT;-Context: competitive sporting environment of any level (local to international).

This scoping review included the academic literature of any study design or publication type. No restrictions were applied to the date or geographical location to facilitate a comprehensive review. Studies were restricted to the English language given the resourcing constraints.

### 2.4. Search Strategy

Test searches were undertaken in Medline via Ovid, Embase, and CINAHL Complete to identify keywords and subject headings were mapped to the relevant results. A librarian assisted with the final search strategy using the indexed keywords and MESH terms. This was then adapted for Medline via Ovid, Scopus, Emcare, SportDiscus, Web of Science, and CINAHL. An example for the range of pelvic floor disorders searched is “Pelvic Floor” OR “urinary incontinence” OR “stress urinary incontinence” OR “urge urinary incontinence” OR “Fecal Incontinence” OR “pelvic organ prolapse” OR cystocele OR “rectal prolapse” OR “uterine prolapse” OR “visceral prolapse” OR “rectal prolapse” OR rectocele OR “Pelvic Floor Disorders” OR “Pelvic dysfunction” OR incontinen*. The complete examples of these are included in Appendix A. The reference lists of studies included in this scoping review were also searched for the additional relevant literature.

### 2.5. Study Selection

After the search was completed, all results were imported into Covidence ™. In Covidence, duplicates were automatically removed and then a two-step screening process of title and abstract screening and full-text screening was undertaken. Studies were excluded at each step if it became clear they did not meet the eligibility criteria. Both steps involved independent screening of the articles by two authors. If there was a conflict, this was resolved via discussion or via review from a third author.

Articles were not systematically or critically appraised, in line with the JBI methodology [16]. Study selection was reported in the Preferred Reporting Items for Systematic Reviews and Meta-analyses (PRISMA 2020) flow diagram [17].

### 2.6. Data Extraction

The extracted data included study author/s, publication year, study design, participant characteristics, study objective, main findings, and definitions of ‘PFD’ and ‘PFMT’. The main findings reported were only those relevant to the topic of this review, consistent with the Johanna Briggs Institute scoping manual. The authors developed a data extraction form that was designed and refined over multiple team meetings. The data were analyzed for content [18], and the main findings were synthesized under the subheadings of ‘athletes’ knowledge of PFD’, ‘athletes’ knowledge of PFMT’, ‘athletes’ attitudes toward PFMT’, and ‘athletes’ practice of PFMT’.

## 3. Results

As shown in Figure 1, 1021 articles were identified and imported into Covidence. A total of 35 papers met our eligibility criteria. There was one instance where a conference abstract described an already-included full-text study in a different format [19,20]; therefore, the sample includes 35 papers describing 34 studies.

### 3.1. Characteristics of the Included Studies

Of the 35 included publications, 29 were primary research articles (4 were abstracts), with 23 studies being quantitative and 6 using mixed methods. The sample included three systematic review articles, the two literature reviews, and one narrative review. The oldest study was published in 2002; however, more than half of the studies were published in the last four years.

### 3.2. Characteristics of the Participants

The number of participants included in the studies ranged from 11 to 480 athletes, and the largest systematic review included over 2000 participants. Only two quantitative studies reported findings on male athletes, and the remaining literature was specific to females. The parity of athletes was often incompletely reported or absent. Therefore, this review categorized athletes as either having ‘had previous pregnancy or birth’ or ‘no stated previous pregnancy or birth’ as there were insufficient findings to categorize athletes as nulliparous or not across all studies. The majority of women were in the ‘no previous pregnancy or birth’ category. Participant characteristics are presented in Table 1.

Across the 35 papers, there were 241 sports mentioned. The sports mentioned the most frequently were running-related sports (*n* = 24), football/futsal/soccer (*n* = 14), volleyball (*n* = 12), gymnastics (*n* = 12), basketball (*n* = 11), CrossFit (*n* = 11), and weightlifting (*n* = 10). Sports were only counted once from each paper, for example, when a systematic review included three papers on gymnasts, gymnast was only counted once. When considering the level of competition, there were 12 mentions of national-level athletes and 10 mentions, respectively, of both international and local-level athletes; however, many of the studies did not report the competition level. Ultimately, there were too many sports and variable competition levels to comment on the findings between or across sports. Appendix A contains information on sports and competition levels within the sample.

### 3.3. Working Definition of PFD

Only one study defined PFD specifically, stating it was ‘*a collection of signs, symptoms, and conditions that affect the pelvic floor*’, including urinary incontinence (UI), stress UI (SUI), urge UI (UUI), anorectal dysfunction, sexual dysfunction, pelvic organ prolapse (POP), and pelvic pain [4]. Three other studies mentioned the term PFD and some additional conditions may include anal incontinence (AI), lumbopelvic pain, and fecal incontinence (FI) [2,21,22]. The remaining studies either defined or only mentioned components of PFD, including two that were not mentioned in the above five studies: mixed UI (MUI) and athletic incontinence.

All studies defined or mentioned at least one form of UI. Definitions given for the terms AI, FI, and anorectal dysfunction appeared synonymous with each other and these terms were mentioned in eight studies [2,4,5,21,22,23,24,25]. Six studies mentioned POP [2,4,5,21,22,24] and only three defined POP [5,21,22]. Definitions of POP focused on the descent of female pelvic organs [5,21,22]. Appendix A can be referred to for the extracted definitions of terms.

### 3.4. Working Definition of PFMT

The term PFMT was defined in two studies. Jacome et al. [26] suggested ‘*PFMT* … *involves strengthening of the PFM*’ and the second study by Joseph et al. [27] suggested ‘*the training consists typically of regular contraction and relaxation of the pelvic muscles, also known as Kegel exercises*’. Other papers used synonymous terms including pelvic floor muscle exercises (PFME), physiotherapeutic care, and pelvic floor training program; however, these terms were not specially defined. Table 2 outlines further details.

**Table 1 ijerph-22-00104-t001:** Participant characteristics.

Study Details—Author/s, Date	Participants—Number and Age	Sex (*n* = Number of Each Sex)	No Stated Previous Pregnancy or Birth (*n* = Number of Females)	Had Previous Pregnancy or Birth (*n* = Number of Females)	Pregnancy/Birth History Unclear (*n* = Number of Females)	Number of Different Sports Reported by Each Study	Level of Competition
Almousa and Van Loon [28]	2459 aged 12 to 45 years.	2459 F	2459			36	Local, National, International.
Bo and Backe-Hansen [23]	31 elite athletes aged 28 to 35 years, 46 age-matched controls.	31 F			Greater than or equal to 13.	2	National.
Bo and Nygaard [24]	Not stated.	F			All.	10	Local, National, International.
Brennand et al. [29]	59 aged 20 to 67 years.	59 F	10	49		32	Not stated.
Campbell et al. [30]	11 with mean age of 47.6 (S.D. of 9.8).	11 F	2	9		6	Not stated.
Cardoso et al. [31]	118 aged 18–30 years.	118 F	118			6	Not stated.
Carls [19]	86 aged 14–21 years.	86 F			86	7	Not stated.
Carls [20]	171 aged 17 to 21 years.	171 F	171			7	Not stated.
Culleton-Quinn et al. [4]	Not stated.		Participants from 15 studies were nulliparous.		Not specified.	37	Local, National, International
Gan and Smith [32]	Not stated.	F			Not specified.	Not stated.	Not stated.
Garrington et al. [21]	141 athletes Aged 18 years+.	141 F		141		1	Local.
Gill et al. [33]	176 aged 18 to 50 years.		81	88	7	1	Not stated.
Gram and Bo [8]	107 mean age 24.5 years (S.D. 1.6 years).	107 F	107			1	National, International.
Hazar [34]	16 aged 18 to 27 years.				16	1	Not stated.
High et al. [22]	314 aged 20 to 71 years.	314 F		180	134	1	Not stated.
Jacome et al. [26]	106 with a mean age of 23 years (S.D. 4.4 years).	106 F	96	10		6	Local.
Joseph et al. [27]	>2348 aged 18 to 44 years.	>2348 F			Unclear. Aim was to discuss nulliparous athletes, but some studies include parous athletes.	7	Not stated.
Krnicar et al. [35]	28 aged 18 to 23 years.	28 F			28	1	Not stated.
Lúovíksdóttir et al. [36]	18 sportswomen, 16 untrained women all aged 18 to 30 years.	18 F			18	11	Not stated.
Mahoney et al. [37]	425 aged 18 to 69 years.	425 F			425	4	Not stated.
Moreno et al. [38]	120 mean age of 27.2 years (S.D. 5.4 years).	120 F			120	28	International.
Neels et al. [39]	28 aged 14 to 25 years.	28 F	28			Not stated.	Local.
Parmigiano et al. [40]	148 mean age 15.4 years (S.D. 2.0 years).	148 F	148			8	National, International.
Pereira et al. [41]	189 with a mean age of 30.07 years.	189 F	147	42		1	Not stated.
Rohde et al. [42]	342 aged 18 to 65 years.	342 F			342	3	Not stated.
Rolli and Frigeri [43]	120 (60 athletes, 60 non-athletes), mean 19.6 years.	60 F			60	1	Not stated.
Skaug et al. [10]	319 aged 12 to 36 years.	319 F			319	3	National
Skaug et al. [5]	384	180 F, 204 M		49		2	National, International
Stickley and McDowell [11]	279 aged 18 to 26 years.	279 F			279	10	Local.
Thyssen et al. [44]	291 aged 14–51 years.	291 F	278	13		8	National.
Toprak Celenay et al. [2]	88 aged 18 to 32 years.	29 F, 59 M	29			3	Not stated.
Wikander et al. [45]	134 aged 20 to 59 years.	134 F			134	1	Not stated.
Wikander et al. [46]	480 aged 20 to 71 years	480	307	173		1	Local, National, International.
Wikander et al. [47]	191 mean age of 35.92 years (S.D. 12 years)	191 F	119	72		1	Local, National, International.
Wikander et al. [48]	452 aged 20 to 63 years.	452 F			Not specified.	1	Local, National, International.

**Table 2 ijerph-22-00104-t002:** Definitions and terms of PFD and PFMT from included papers.

Study Details—Author/s, Date	Defined and Mentioned Components of PFD	PFMT Definition/Term Used
Almousa and Van Loon [28]	Defined UI, SUI, and UUI. MUI mentioned.	Not defined. Pelvic floor muscle exercises.
Bo and Backe-Hansen [23]	UI, SUI, UUI, and Faecal incontinence (FI) defined. Mixed incontinence mentioned	Not defined. PFMT.
Bo and Nygaard [24]	UI defined. SUI, AI, and POP mentioned.	Not defined. Train the PFM.
Brennand et al. [29]	UI and SUI mentioned.	Not defined. Pelvic floor exercise, Pelvic floor physiotherapy
Campbell et al. [30]	UI, SUI, UUI, and MUI defined.	Not defined. PFMT.
Cardoso et al. [31]	UI defined. SUI, UUI, and MUI mentioned.	Not defined. Strengthening pelvic floor muscles and physiotherapeutic care.
Carls [19]	UI, SUI, and UUI mentioned.	Not defined. Pelvic muscle exercises, Kegals.
Carls [20]	UI, SUI, and UUI mentioned.	Not defined. Pelvic muscle exercises, Kegals.
Culleton-Quinn et al. [4]	PFD, UI, SUI, and UUI defined. Anorectal dysfunction, sexual dysfunction, POP, and pelvic pain mentioned.	Not defined. PFME and PFMT
Pereira et al. [41]	UI and SUI defined. UUI and MUI mentioned.	Not defined. Strengthening pelvic floor muscles and physiotherapeutic care.
Gan and Smith [32]	UI, SUI, and UUI mentioned.	Not defined. Pelvic floor physical therapy, Pelvic floor muscle physiotherapy
Garrington et al. [21]	PFD mentioned. UI, AI, and POP defined.	Not defined. PFMT.
Gill et al. [33]	UI, SUI, and UUI mentioned.	Not defined. Pelvic floor muscle exercises.
Gram, and Bo [8]	Defined UI and SUI. UUI and MUI mentioned.	Not defined. PFMT.
Hazar [34]	Defined UI.	Not defined. Not referred to.
High et al. [22]	PFD mentioned. General UI, FI, UUI, SUI, and Symptomatic POP defined.	Not defined. Kegals, Pelvic floor physical therapy.
Jacome et al. [26]	UI defined. SUI, UUI, and MUI mentioned.	‘…PFMT, which involves strengthening of the pelvic floor muscles.’
Joseph et al. [27]	Defined UI and SUI.	PFMT. The training consists typically of regular contraction and relaxation of the pelvic muscles, also known as Kegal exercises.
Krnicar et al. [35]	SUI mentioned	Not stated. PFMT
Lúovíksdóttir et al. [36]	UI and SUI defined.	Not defined. Pelvic floor exercises.
Mahoney et al. [37]	SUI defined.	Not defined. Pelvic floor physical therapy, Kegal exercises, pelvic floor training
Moreno et al. [38]	SUI defined. UI mentioned.	Not defined. Pelvic floor strengthening exercises.
Neels et al. [39]	UI and SUI mentioned.	Not defined. PFMT. Pelvic floor exercises.
Parmigiano et al. [40]	UI mentioned.	Not defined. Not referred to.
Rohde et al. [42]	SUI defined.	Not defined. PFMT.
Rolli and Frigeri [43]	Mentioned UI, SUI, and UUI.	Not defined. Pelvic floor exercises, pelvic floor training.
Skaug et al. [10]	SUI and AI defined. UI mentioned.	Not defined. PFMT.
Skaug et al. [5]	UI, AI, and POP defined.	Not defined. PFMT.
Stickley and McDowell [11]	UI, SUI, UUI, and MUI defined.	Not defined. Pelvic floor physical therapy, pelvic floor exercises.
Thyssen et al. [44]	UI mentioned.	Not defined. Pelvic floor training program.
Toprak Celenay et al. [2]	PFD, UI, SUI, UUI, AI, POP, lumbo-pelvic pain, and sexual dysfunction mentioned.	Not defined. PFME
Wikander et al. [45]	UI defined. SUI mentioned.	Not defined. Pelvic floor exercises.
Wikander et al. [46]	UI, SUI, UUI, MUI, and athletic incontinence defined	Not defined. Pelvic floor exercises
Wikander et al. [47]	UI, SUI, UUI, and MUI defined.	Not defined. Pelvic floor exercises.
Wikander et al. [48]	Athletic incontinence defined. UI and SUI mentioned.	Not defined. Pelvic floor exercises.

### 3.5. Athletes’ Knowledge of PFD

Eighteen studies discussed athletes’ knowledge of PFD [2,4,5,8,21,24,25,26,28,31,32,34,36,37,38,40,41,43]. Thirteen of these were quantitative studies, three were systematic reviews, and two were literature reviews. The common findings were that many athletes knew little about the normal function of the PFM [2,4,5,8,21,25,28,36,43].

Several studies found gaps in athletes’ knowledge of UI specifically [26,28,32,34,37,38,40,41]. Two studies assessed and graded athletes’ knowledge as adequate or inadequate using the Knowledge, Attitudes, and Practice (KAP) survey created by Cardoso et al. (2018). According to the KAP, only 53.4% of female cross-fitters and 31% of high-impact athletes had adequate knowledge [31,41]. Both studies including male athletes found that men had less PF knowledge than women [2,5].

### 3.6. Athletes’ Knowledge of PFMT

Seventeen papers discussed athletes’ knowledge of PFMT [2,4,5,8,19,20,24,25,26,27,28,29,30,31,32,35,41]. Ten of these were quantitative, one used mixed methods, two were systematic reviews, and three were literature reviews. Many athletes did not know how to train the PFM [4,5,8,24,25] and many also did not know why they should undertake the training [4,5,8,25]. Males knew less than females in relation to both of these questions [5].

Several athletes had not heard of PFMT [2,20,27,28,30], and even if they had, some were unaware it could be used to treat PFD [19,26,29,32]. Similarly, Cardoso et al. (2018)’s KAP survey determined that many athletes’ knowledge of PFMT was poor. They also found that athletes with adequate knowledge of PFD and PFMT were 57% less likely to develop UI [31]. A higher number of athletes demonstrated adequate knowledge in 36. Pereira et al.’s [41] study, which reported that knowledge was not significantly associated with UI. Krnicar et al. [35] reported the most positive findings, with 85.71% of the respondents demonstrating a familiarity with correct pelvic floor muscle contractions.

### 3.7. Athletes’ Attitudes Toward PFMT

Fifteen papers discussed athletes’ attitudes toward PFMT [4,5,11,19,20,25,29,30,31,39,41,42,46,47,48]. Eleven of these were quantitative, four used mixed methods, and one was a systematic review.

Several studies reported that athletes would use PFMT to treat PFD if they knew how [4,5,11,20,25]. Other athletes conveyed interest in receiving education regarding PFMT and treatment via pelvic floor physiotherapy [19,29,39,42]. Some athletes gave the caveat that they would only use PFMT if UI became severely bothersome [42]. Another study that reported on athletes describing the tailored physiotherapy intervention they received (which included PFMT) reported that the athletes viewed the PFMT as valuable [30]. According to the KAP, athletes’ attitudes toward PFMT were mostly adequate [31,41].

Once aware of PFMT, athletes’ confidence in their ability to perform PFMT was reasonably high, with studies reporting that close to 70% of female athletes were confident, regardless of whether they had PFD or not [4,46,47,48].

### 3.8. Athletes’ Practice of PFMT

Fifteen studies discussed athletes’ practice of PFMT [2,4,22,23,25,31,33,35,36,37,41,44,45,47,48]. Eleven of these were quantitative, three used mixed methods, and one was a systematic review. In 12 studies, athletes reported practicing PFMT at some point [2,4,22,23,25,33,35,37,44,45,47,48]. However, the practice of PFMT was generally low, with many studies reporting that less than 10% of participants had practiced PFMT [2,4,25,37,44,45]. In one study including male athletes, four males and four females reported practicing PFMT [2]. However, as the overall sample in this study included 29 females and 59 males, females were concluded as more likely to practice PFMT than males [2]. Similarly, Lúovíksdóttir et al. [36] determined that female athletes were more likely to practice PFMT than non-athletic female controls.

Of those who did practice PFMT, many reported using it to manage UI [4,33,37,44,45,47,48].

Using the KAP survey, Pereira et al. [41] determined that 3.7% of athletes had adequate practice of prevention, management, and treatment strategies, whereas Cardoso et al. [31] found that 0% had adequate practice. Table 3. provides a summary of the outcomes measured in the included studies and Table 4. provides details on the outcomes themselves.

## 4. Discussion

This scoping review has collated the available evidence on athletes’ knowledge of PFD and knowledge of and engagement with PFMT. Additionally, the review has mapped the existing literature to establish the working definitions of ‘PFD’ and ‘PFMT’ and explored the types of athletes that have been studied. The main findings were that athletes had low practice of PFMT, and poor knowledge of PFD and PFMT, although most athletes expressed positive attitudes towards PFMT. The working definitions of PFD and PFMT are ill-defined, and the researched athletic population is too heterogeneous (both in sport type and competition level) to confidently comment on whether findings were consistent between or across sports. The literature is lacking describing PFMT in male athletes.

The literature on athletes’ knowledge of PFD and PFMT is sparse but emerging, with most included studies being published in the last four years. Since this review was completed, another study was published regarding female athletes’ knowledge of PFD [50]. Bosch-Donate et al. [50] found that athletes’ knowledge was generally poor, except for knowledge of UI, where more than 70% of participants provided correct responses [50]. This finding makes sense in the context of the current review and the extant literature, as UI is the most researched and well-defined component of PFD [3].

The high proportion of studies identified in this review focusing only on women may derive from the higher prevalence of PFD amongst females compared to males [5]. The recent increase in publications specific to female pelvic floor function demonstrates efforts to increase the literature focusing on women’s healthcare and women in sports and efforts to improve sex and gender equity [51].

Two of the publications included in this review considered the relationship between knowledge and the prevalence of PFD [31,41]. Cardoso et al. [31] found a significant relationship between knowledge and PFD, stating that those with adequate knowledge were 57% less likely to have PFD. This suggests that educating athletes on this topic may decrease the prevalence of PFD. This idea is supported by studies suggesting that athletes would complete PFMT if they knew how [4,5,11,20,25]. Additionally, athletes in Carls’ [19] study responded affirmatively to wanting pelvic floor education, and similarly, athletes in Campbell et al.’s [30] study described PFMT education as valuable.

There are, however, several factors to consider before suggesting education as an intervention. For instance, the findings in Pereira et al.’s 2022 study [41] contrasted with Cardoso et al. [31] as they found no significant relationship between knowledge and PFD. Subsequently, improving athletes’ knowledge may not produce the desired outcome of decreasing PFD. Pereira et al. [41] found that even when athletes had positive attitudes toward PFMT, they still did not have adequate practice of PFMT, suggesting that variables other than knowledge and attitude influence practice. While Campbell et al. [30] educated athletes, their study was not designed to test the effect of this education on knowledge or engagement with PFMT, which means their work does not prove that education will improve athletes’ knowledge and engagement with PFMT. However, their qualitative findings suggest that future research on this intervention is warranted [30]. Education as an intervention is further supported in a recent review that described the potential for education to contribute to the rehabilitation of dysfunction in athletes [52].

This paper should encourage health professionals to consider the risks associated with athletes with a low knowledge of PFD and PFMT and low engagement with PFMT, and what their role may be in mitigating these risks. One role may be working with athletes and coaching in improving awareness of pelvic floor muscle training, or instigating screening for people who exercise [53]. Future research should explore the components of successful educational interventions and consider investigating whether the results of this review are relevant to male athletes. Components of PFD other than UI also warrant further research.

### Limitations of This Review

The methodology of this review retrieved a broad range of studies and the heterogeneity of qualitative data made it difficult to synthesize findings. There were very few components of PFD other than UI explored and it is unclear whether the results would have been the same if more publications explored AI, POP, or other forms of PFD.

## 5. Conclusions

Despite positive attitudes toward PFMT, the athletes identified in this review had low practice of PFMT and poor knowledge of PFD and PFMT. The current review also identified that health professionals may have a role in addressing these deficits through educational interventions. The concepts of PFD and PFMT are ill-defined, which makes comparisons across the included studies difficult. A wide range of sport types and competition levels across the athletes limit the generalizability of findings. One publication identified in the review reported that higher levels of knowledge were associated with a lower prevalence of PFD, indicating that the education of athletes may have a role in supporting pelvic floor health. Educational interventions for athletes warrant further investigation.

## Figures and Tables

**Figure 1 ijerph-22-00104-f001:**
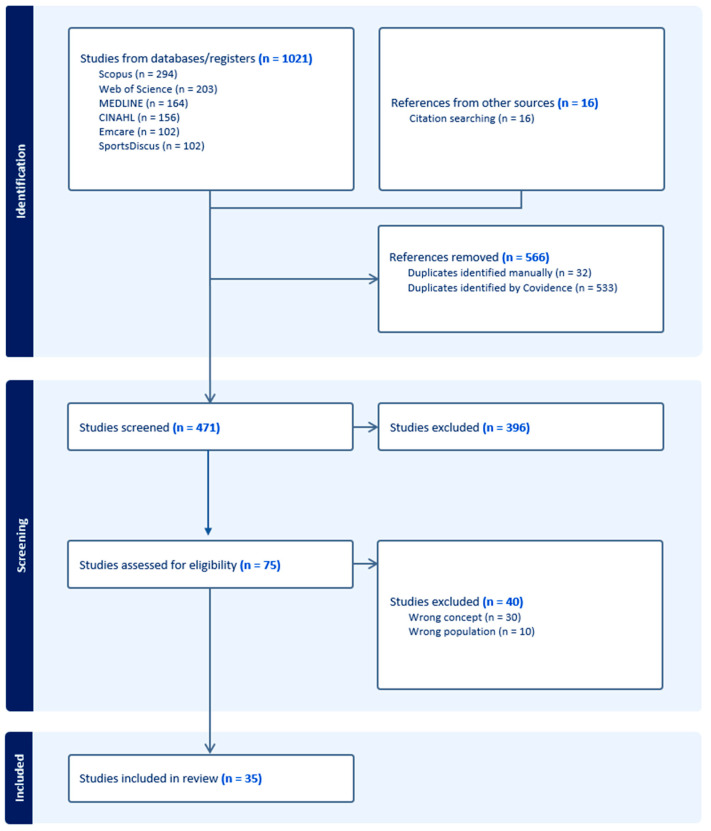
Preferred reporting items for systematic reviews and meta-analyses 2020.

**Table 3 ijerph-22-00104-t003:** Summary of the findings each study reported on.

Study Details—Author/s, Date	Design	Knowledge PFD	Knowledge PFMT	Attitudes PFMT	Practice PFMT
Almousa and Van Loon [28]	Systematic Review of 23 original primary studies.	Y	Y	N	N
Bo and Backe-Hansen [23]	Quantitative.	N	N	N	Y
Bo and Nygaard [24]	Narrative Review.	Y	Y	N	N
Brennand et al. [29]	Quantitative.	N	Y	Y	N
Campbell et al. [30]	Mixed Methods.	N	N	Y	N
Cardoso et al. [31]	Quantitative.	Y	Y	Y	Y
Carls [19]	Quantitative—only research poster abstract available.	N	Y	Y	N
Carls [20]	Quantitative.	N	Y	Y	N
Culleton-Quinn et al. [4]	Systematic Review of 32 primary studies.	Y	Y	Y	Y
Pereira et al. [41]	Quantitative.	Y	Y	Y	Y
Gan and Smith [32]	Literature review.	Y	Y	N	N
Garrington et al. [21]	Systematic review of 8 primary studies, 2 of the studies were on athletes.	Y	N	N	N
Gill et al. [33]	Quantitative.	N	N	N	Y
Gram, and Bo [8]	Quantitative.	Y	Y	N	N
Hazar [34]	Quantitative.	Y	N	N	N
High et al. [22]	Quantitative.	N	N	N	Y
Jacome et al. [26]	Quantitative.	Y	Y	N	N
Joseph et al. [27]	Literature review of 52 studies.	N	Y	N	N
Krnicar et al. [35]	Mixed Methods.	N	Y	N	Y
Lúovíksdóttir et al. [36]	Quantitative.	Y	N	N	Y
Mahoney et al. [37]	Quantitative.	Y	N	N	Y
Moreno et al. [38]	Quantitative.	Y	N	N	N
Neels et al. [39]	Mixed-Methods—only abstract available.	N	N	Y	N
Parmigiano et al. [40]	Quantitative.	Y	N	N	N
Rohde et al. [42]	Mixed methods—only abstract available.	N	N	Y	N
Rolli and Frigeri [43]	Quantitative.	Y	N	N	N
Skaug et al. [10]	Quantitative.	Y	Y	Y	N
Skaug et al. [5]	Quantitative.	Y	Y	Y	Y
Stickley and McDowell [11]	Quantitative.	N	Y	Y	N
Thyssen et al. [44]	Quantitative.	N	N	N	Y
Toprak Celenay et al. [2]	Quantitative.	Y	Y	N	Y
Wikander et al. [45]	Mixed Methods.	N	N	N	Y
Wikander et al. [23]	Quantitative.	N	N	Y	N
Wikander et al. [47]	Mixed Methods.	N	N	Y	Y
Wikander et al. [48]	Quantitative.	N	N	Y	Y

Table Key: Y = reported on. N = not reported on.

**Table 4 ijerph-22-00104-t004:** Knowledge of PFD and PFMT, attitudes and practice of PFMT.

Study Details—Author/s, Date	Objective	Findings—Knowledge of PFD	Findings—Knowledge of PFMT	Findings—Attitudes Towards PFMT	Findings—Practice of PFMT (Habitual Application of PFMT)
Almousa and Van Loon [28]	Primarily to systematically review studies investigating the prevalence of UI in nulliparous female athletes; and secondary, to explore female athletes’ knowledge of and attitudes toward UI.	One study reported 89.9% of the athletes were not familiar with the occurrence of UI as a condition [40], another that 73.3% were unfamiliar with UI and the function of the pelvic floor [43], and two others that 80.8–91% had either not received any pelvic floor education or had never heard of pelvic floor muscle exercises [20,21].	80.8–91% had either not received any pelvic floor education or had never heard of pelvic floor muscle exercises [20,21].	No findings.	No findings.
Bo and Backe-Hansen [23]	The primary aim was to study prevalence of low back pain, pelvic girdle pain, and UI and FI before, during, and after pregnancy in elite athletes and age-matched controls. Secondary aims were to investigate physical activity levels during and after pregnancy, birth history, birth complications, and birth weight of children in the same two groups.	No findings.	No findings.	No findings.	19.4% of athletes were practicing PFMT the year before pregnancy, 14.8% were practicing during pregnancy, 51.7% were practicing at 6 weeks postpartum, and 35.5% were practicing at the time of this questionnaire. These values were not statistically different compared to age-matched controls.
Bo and Nygaard [24]	To describe and discuss the evidence supporting or refuting two hypotheses: 1. General exercise training strengthens the pelvic floor 2. General exercise training overloads, stretches, and weakens the pelvic floor. As well as how exercise influences PFM strength, muscle fatigue, pelvic floor morphology, pelvic floor disorders, and labour and birth variables.	Young, nulliparous women in general, and athletes in particular, have low level of knowledge about the pelvic floor [31].	Young, nulliparous women in general, and athletes in particular, have low level of knowledge about the pelvic floor and little knowledge about how to train the PFM [31].	No findings.	No findings.
Brennand et al. [29]	To identify which specific activities cause leakage and describe the associated severity, exploring how urinary leakage affects physical activity levels, characterizing adaptive mechanisms women use to counteract leakage, and describing the interest a cohort of physically active women experiencing exercise-related incontinence have in receiving treatment for urine loss.	No findings.	Prior to engaging in this study, one-third of women did not know treatment was available.	Most (88.1%) respondents expressed interest in receiving treatment. The greatest interest was in receiving PF physical therapy (84.6%).	No findings.
Campbell et al. [30]	To investigate the feasibility and acceptability of conducting a future trial of physiotherapy to manage UI in athletic women.	No findings.	No findings.	“All those interviewed described the intervention [up to seven sessions of tailored physiotherapy delivered over 6-months, with all programs including elements of PFMT] as ’valuable’. The tailored advice and progression were particularly regarded as a positive point…”	No findings.
Cardoso et al. [31]	To evaluate the prevalence of UI in female athletes practising high-impact sports and its association with knowledge, attitude and practice.	31% of athletes demonstrated adequate knowledge.	31% of athletes demonstrated adequate knowledge. Those with adequate knowledge demonstrated a 57% lower chance of developing UI.	53% of athletes demonstrated adequate attitude.	0% of athletes demonstrated adequate practice.
Carls [19]	To identify the prevalence of SUI and a needs assessment for preventative UI education.	No findings.	91% had never heard of PFM exercises as a means for preventing UI.	83% of those with SUI and 74% of those without SUI responded affirmatively to wanting PFMT education.	No findings.
Carls [20]	To assess the prevalence of UI in young female athletes and to determine educational needs of the athletes regarding prevention and treatment of UI.	No findings.	91% of athletes had never heard of pelvic floor muscle exercises or Kegal exercises.	83% of athletes with SUI and 74% of those without symptoms reported that they would try pelvic floor muscle exercises if they knew how to perform them properly.	No findings.
Culleton-Quinn et al. [4]	To investigate the experiences of symptoms of PFD in elite female athletes.	41.4% of women in one study [25] had never heard about the PFM.	In one study, 42.8% of women did not know why and 44.4% did not know how to train PFM [5]. In another study 10% of women knew how and 18.2% of women knew why to train the PFM [25].	In one study, 78.3% of women reported they would do PFMT if they knew how [5]. 73.7% of women in another study reported they would do PFMT to prevent or treat PFD if they knew how [25]. In another study, 26% of women reported they were not confident in their ability to perform PF exercises correctly [47]. In another study, 71.71% of participants stated they were confident or very confident regarding PF exercises [48]. In another study, 77.1% of participants said they were confident or very confident in their ability to perform PF exercises [33].	PFMT was a strategy used by participants in 7/23 studies. 0.9% of women in one study reported they had tried PFMT [25]. 4.6% of women in another study had completed PFMT because of UI [49]. 7.4% of women in another study reported using PF exercises to control or minimise UI [44].
Gan and Smith [32]	To summarize an understanding of the epidemiology, pathophysiology, and management strategies of UI in female athletes.	One study found that athletes were not aware of the link between sports and UI [26].	One study found that athletes were not aware of the methods to prevent or treat UI [26].	No findings.	No findings.
Garrington et al. [21]	To determine the current extent to which specific prevention and management interventions for UI, AI, and POP have been studied in female military and athlete populations and how effective and safe these are.	One study found that 80.8% of participants were not aware of the role of PFMs [50].	No findings.	No findings.	No findings.
Gill et al. [33]	Primary aim was to establish the prevalence of UI among netball players within a rural netball league in South Australia. Secondary aims were to establish prevalence of sub-types of UI, severity and bother of UI, and self-management strategies in this cohort.	No findings.	No findings.	No findings.	17 athletes (31.5% of those with UI) reported using PFMT as a self-management strategy for their UI.
Gram, and Bo [8]	To investigate prevalence and risk factors of UI and investigate the impact of UI on performance and their knowledge of the pelvic floor and PFMT.	69.1% had never heard about the pelvic floor.	73.9% did not know why, and 77.6% did not know how they should train the pelvic floor muscles.	No findings.	No findings.
Hazar [34]	To investigate the UI in female volleyball players.	Almost all of the female volleyball players (90.9%) did not know that UI is a worldwide problem in women.	No findings.	No findings.	No findings.
High et al. [22]	To estimate the prevalence of pelvic floor disorders by symptoms in female CrossFit athletes in the United States and characterize subjects reporting POP symptoms, UI, and FI.	No findings.	No findings.	No findings.	21% of athletes had tried Kegals in the past, 18% were using Kegals currently, and 7% reported use of pelvic floor physical therapy at some point.
Jacome et al. [26]	To assess the prevalence of UI in a group of female athletes and to explore its impact on their lives.	Athletes considered urine loss to be a normal condition that is unrelated to practicing sports.	The athletes were not aware of methods to prevent or treat the condition.	No findings.	No findings.
Joseph et al. [27]	To assess the prevalence of SUI and explore how it is associated with different sports. Also aimed to review the currently available treatments and the impact SUI may have on athletes’ well-being.	No findings.	One study found that a group of gymnasts were not aware of PFMT [8].	No findings.	No findings.
Krnicar et al. [35]	To analyse how common stress urinary incontinence is between the ages of 18 and 23 and to find out how much PFMT is included in high-intensity swimming training.	No findings.	85.71% of the respondents expressed a familiarity with correct pelvic floor muscle contractions.	No findings.	PFMT was rarely or occasionally used in the training regime.
Lúovíksdóttir et al. [36]	To examine pelvic floor muscle strength, incontinence, and women’s knowledge of pelvic floor muscle tension among athletic and non-trained women.	Four sportswomen said they thought little and knew little about the PFM.	No findings.	No findings.	The female athletes were more likely to do PFMT than the untrained women.
Mahoney et al. [37]	To add to the body of research on incontinence specific to female strength athletes by exploring prevalence of SUI, normalization of SUI, preferences on sources for information on SUI, and rates of seeking treatment for SUI in female strength athletes participating in weightlifting.	64.9% of athletes stated that they think UI is a normal part of their sport.	No findings.	No findings.	8.5% of athletes had tried pelvic floor physical therapy for SUI.
Moreno et al. [38]	To survey female Brazilian Olympic athletes prior to the Summer Olympic Games held in Rio de Janeiro, 2016, to identify the professionals who they sought for follow-up. In addition, it aimed to investigate the athletes’ knowledge regarding gynaecological issues related to sports practice: female athlete triad, athletic urinary incontinence, and weight control concerns, and lastly, to relate the multidisciplinary approach to their knowledge.	67.5% were aware that involuntary urine loss could happen during sports practice.	No findings.	No findings.	No findings.
Neels et al. [39]	To estimate the prevalence of UI in young adult female athletes; to introduce basic education about the pelvic floor and pelvic floor exercises in their training program; and to evaluate how this study is perceived by the participants and its influence on UI.	No findings.	No findings.	89.3% declared to be interested to learn the pelvic floor exercises.	No findings.
Parmigiano et al. [40]	To propose the inclusion of the gynaecological investigation during the pre-participation assessment of women who engage in physical exercise, using a specific tool called Pre-Participation Gynaecological Examination (PPGE).	89% are not aware ofthe possibility of involuntary urine loss in the populationof athletes.	No findings.	No findings.	No findings.
Pereira et al. [41]	To determine the prevalence and factors associated with UI in female Cross fitters.	53.4% of participants had adequate knowledge about UI.	53.4% of athletes had adequate knowledge.	86.2% of athletes had adequate attitudes towards PFMT.	Only 3.7% of athletes had adequate practice of prevention, management and treatment.
Rohde et al. [42]	To identify the scope of the problem of SUI in women participating in strength sports, movement patterns that elicit SUI, and the impact on quality of life in female athletes with SUI.	No findings.	No findings.	Women stated that if provided education about PFMT they would use it but only once the urinary leakage episodes became severely bothersome.	No findings.
Rolli and Frigeri [43]	To estimate the real prevalence of symptoms associated with UI in nulliparous female basketball players in comparison to nulliparous non-athletes.	Of the symptomatic athletes, 73.3% were not aware of the function or importance of the pelvic floor.	No findings.	No findings.	No findings.
Skaug et al. [10]	To investigate the prevalence and risk factors for UI and AI in female artistic gymnasts, team gymnasts, and cheerleaders; the influence of UI and AI on daily living and sports performance; and the athletes’ knowledge about the PFM.	One hundred thirty-two (41.4%) of the athletes had never heard about the PFM. In total, 39 (12.2%) of the athletes reported that they had heard about the PFM from their coach, 32 (10.0%) from teammates, 61 (19.1%) from health personnel, and 54 (16.9%) from other sources (friends, siblings or parents). The mean self-rated knowledge of the PFM was 1.5 (SD: 1.7) of 10.	In total, 32 (10.0%) knew how and 58 (18.2%) why to train the PFM.	Two hundred thirty-five (73.7%) responded they would do PFM training to prevent or treat UI and AI if they knew how.	Three athletes (0.9%) reported they did or had tried PFM training.
Skaug et al. [5]	To investigate the prevalence and risk factors for PFD in powerlifters and Olympic weightlifters. Furthermore, to investigate the impact and bother of PFD and knowledge of the PFM among the same athletes.	In total, 37 women (20.6%) and 120 men (58.8%) had never heard about PFM.	In total, 77 (42.8%) women and 150 (73.5%) men did not know why, and 80 women (44.4%) and 148 men (72.5%) did not know how to train the PFM.	141 (78.3%) women and 101 (49.5%) men responded they would do PFM training to prevent or treat PFD if they knew how.	No findings.
Stickley and McDowell [11]	To assess the prevalence of UI among female collegiate athletes and to evaluate the impact of incontinence on individual function and perceived athletic performance, and to determine athletes’ awareness of physical therapist management of UI.		88.5% of participants were unaware that a physical therapist could treat UI. 38.5% reported that they had heard of Kegal exercises, and 51.9% reported they had not. 9.6% did not respond to the question about Kegals.	36.5% of participants reported a willingness to try PFMT if they knew how. Overall, 32.7% of athletes said they would not be interested and 30.8% of athletes did not respond to this question.	No findings.
Thyssen et al. [44]	To elaborate on the problem of UI among elite athletes and dancers while participating in their sport and during daily life activities.	No findings.	No findings.	No findings.	6 women (4.6%) had completed a pelvic floor training program because of urine loss.
Toprak Celenay et al. [2]	To evaluate the pelvic floor knowledge and awareness level and the lower urinary tract symptoms of both female and male athletes and to compare the pelvic floor knowledge and awareness levels between genders.	Only 23 athletes (26.1%) had heard of the PFM. Seventy-three athletes (83.0%) could not identify the location of the PFM. Most of the athletes (84.1%) did not identify any PFM function. The pelvic floor knowledge level was higher in female athletes than in male athletes (*p* < 0.05).	84.6% of the athletes had not heard of PFME and 87.5% did not know how to perform them.	No findings.	Most of the athletes (90.9%) had never performed PFME. Overall, 9.1% of the athletes had performed PFME. In total, 4 females and 4 males had performed PFME.
Wikander et al. [45]	To investigate the relationship between commonly cited risk factors and the incidence of UI in competitive women powerlifters.	No findings.	No findings.	No findings.	Two women reported using pelvic floor exercises to control or minimise UI.
Wikander et al. [46]	To determine the prevalence of UI in competitive women powerlifters; identify possible risk factors and activities likely to provoke UI; and establish self-care practices.	No findings.	No findings.	Most women reported being confident (46.5%) or very confident (25.2%) in their ability to correctly perform a pelvic floor contraction. In the subset of women who had experienced UI at some point in their life, 47.4% reported being confident in their ability to perform pelvic floor exercises and 19.2% were very confident in their abilities.	Women stated in the survey the strategies utilized to prevent, reduce, or contain leakage. Many self-care strategies revolved around activation of the pelvic floor and the table created by the authors of this study listed 18 different responses that referred to the use of PFMT. However, it is unclear whether only 18 participants stated these strategies or whether the authors summarised the list into 18 points.
Wikander et al. [47]	To explore the multifactorial issue of UI in competitive women weightlifters focusing on prevalence, risk factors, and activities that provoke UI. In addition, to identify self-care strategies used by incontinent competitive women weightlifters and the level of confidence they have in performing pelvic floor exercises and utilising women’s health professionals.	No findings.	No findings.	Overall, 77.1% of incontinent subjects stated that they were either confident or very confident in their ability to perform pelvic floor exercises. In total, 22.9% of the women were not confident in their ability to perform pelvic floor exercises.	Women were asked to state the self-care strategies they used. These are the ones that may indicate practice of PFMT: - Consciously engaging pelvic floor before lifting - Practicing pelvic floor exercises outside training - Engaging in release work and massage, stretching the lower back and hips, and focusing on pelvic mobility - Strengthening deep muscles and core training
Wikander et al. [48]	To determine prevalence of urinary and athletic incontinence and establish which activities and contexts were most likely to provoke urine leakage in women CrossFit competitors.	No findings.	No findings.	Overall, 26% of women who reported UI at some point in their life were not confident in their ability to correctly perform pelvic floor exercises.	No findings.

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
