# Peer review of "Athletes’ Knowledge of Pelvic Floor Dysfunction and Their Knowledge of and Engagement with Pelvic Floor Muscle Training: A Scoping Review"

_ijerph, 2025, doi:10.3390/ijerph22010104_

Round 1

Reviewer 1 Report

Comments and Suggestions for Authors

1. It is important to include coaches, not just athletes, as they are also unaware of this problem and how to prevent it. It is important to include pelvic floor muscle training routines in the daily training routines of different sports. I think it would be interesting to talk about this in the introduction.

2. Why has pubmed not been used?

3. Why was there no selection of a specific sport or sports?

For example, those sports with the highest prevalence. Perhaps the results would have been more specific.

4. Why was there no differentiation between the different levels of practice? Could there be differences between an international level athlete and another athlete at a regional level? And if so, what factors could this be due to?

5. Within the pelvic floor dysfunctions, why was the most prevalent and most closely related to the practice of sport not selected?

6. The term pelvic floor dysfunctions is very broad, it would have been better to focus the search on a specific dysfunction.

7. Have you taken into account sport as a risk factor for some pelvic floor disorders? And within them, which ones have the highest prevalence?

Author Response

REVIEWER ONE

  1. It is important to include coaches, not just athletes, as they are also unaware of this problem and how to prevent it. It is important to include pelvic floor muscle training routines in the daily training routines of different sports. I think it would be interesting to talk about this in the introduction. Agreed, we have added a statement on the lack of training of coaching staff in pelvic floor muscles and the potential that targeting this may have.
  2. Why has pubmed not been used? We searched medline using OVID – this accesses the same data as using pubmed.
  3. Why was there no selection of a specific sport or sports? We did collect and collate the types of sports included in the literature, however many studies reported on many different sports and we were not able to synthesize this is a meaningful way (see Table 1)

For example, those sports with the highest prevalence. Perhaps the results would have been more specific.  We agree

  1. Why was there no differentiation between the different levels of practice? Could there be differences between an international level athlete and another athlete at a regional level? And if so, what factors could this be due to? We agree that this would be interesting to investigate further, however the reporting of the levels of sport participation was not presented in a way that would allow us to make this comparison. Table 1 presents the information on levels of sport participation from the literature.
  2. Within the pelvic floor dysfunctions, why was the most prevalent and most closely related to the practice of sport not selected? The data we collated and reviewed did not allow this level of investigation. Although we agree that this analysis would have been of interest if it were possible with the data that was available.

6. The term pelvic floor dysfunctions is very broad, it would have been better to focus the search on a specific dysfunction. We searched for this term generally as well as specific dysfunctions. We have included an example search strategy to confirm this, and the complete list for all data bases is provided in supplemental file one.  "Pelvic Floor" OR "urinary incontinence" OR "stress urinary incontinence" OR "urge urinary incontinence" OR "Fecal Incontinence" OR "pelvic organ prolapse" OR cystocele OR "rectal prolapse" OR "uterine prolapse" OR "visceral prolapse"  OR rectocele OR "Pelvic Floor Disorders" OR "Pelvic dysfunction" OR incontinen*. Whilst focusing on a specific dysfunction may have produced a more specific set of results for a more focused sample – clinically, it is not common for people to present with one isolated element of pelvic floor dysfunction. Given the common cross over of these conditions, we choose to keep the study broad.

  1. Have you taken into account sport as a risk factor for some pelvic floor disorders? And within them, which ones have the highest prevalence? Thank you for this comment. The data around prevalence is outside of the scope of this paper.

Reviewer 2 Report

Comments and Suggestions for Authors

The report is a scoping review of six databases searched in January 2024 regarding athletes’ knowledge of PFD and PFMT and their engagement with PFMT. The aim was to determine the type of athletes researched.

The strengths are that the work is well-written and generally follows the process for scoping reviews, with the conclusions following from the results. 

The weaknesses are several:

  1. There are no keywords following the Abstract,
  2. The authors have not used the most recent guidelines for scoping reviews: https://www.prisma-statement.org, Peters, M. D. J., Godfrey, C., McInerney, P., Khalil, H., Larsen, P., Marnie, C., Pollock, D., Tricco, A. C., & Munn, Z. (2022). Best practice guidance and reporting items for the development of scoping review protocols. JBI Evidence Synthesis20(4), 953–968.https://doi.org/10.11124/JBIES-21-00242,
  3. The flow diagram is not standard, 
  4. The size of Table 1 is needlessly large, 
  5. The checklist is missing from the Supplementary file,
  6. There are insufficient details offered regarding the scoping review process for it to be reproducible,
  7. No searched keywords are provided for each database searched.
  8. The authors have not demonstrated what their research adds to the literature. Here is a Google Scholar search of the topic for research published since 2020: https://scholar.google.ca/scholar?as_ylo=2020&q=Athletes’+Knowledge+of+Pelvic+Floor+Dysfunction+and+their+2+Knowledge+of+and+Engagement+with+Pelvic+Floor+Muscle+Train-+3+ing:+a+Scoping+Review&hl=en&as_sdt=0,5. There are “About 1,100 results”. The authors must read the most relevant of these and compare and contrast their work.

Additionally, please find supporting citations published since 2020 for those outdated. These include, in the Introduction, citations [6] and [10]. In the Material and Methods, these include [14] (see reference offered above), [15], and [17].

Author Response

REVIEWER TWO

The weaknesses are several:

  1. There are no keywords following the Abstract,

Thank you – these have been added.

  1. The authors have not used the most recent guidelines for scoping reviews: https://www.prisma-statement.org, Peters, M. D. J., Godfrey, C., McInerney, P., Khalil, H., Larsen, P., Marnie, C., Pollock, D., Tricco, A. C., & Munn, Z. (2022). Best practice guidance and reporting items for the development of scoping review protocols. JBI Evidence Synthesis20(4), 953–968.https://doi.org/10.11124/JBIES-21-00242,

Thank you – we have updated this reference and the references suggested below

  1. The flow diagram is not standard, 

This flow diagram was exported from the covidence software program that was used for the search. We note that it has formatted incorrectly in the manuscript and have replaced this.

  1. The size of Table 1 is needlessly large, Thank you – we have reformatted this
  2. The checklist is missing from the Supplementary file, Thank you – we have added this as supplemental file (not for publishing)T
  3. There are insufficient details offered regarding the scoping review process for it to be reproducible, Thank you for your comment. We have reviewed the information provided in the methods section under study selection and consider that this, in conjunction with the search strategies in the supplemental file do allow this review to be reproduced.
  4. No searched keywords are provided for each database searched. A complete set of search terms are provided in supplementary file one.  We have added an example to the paper to improve the reproducibility of this review.
  5. The authors have not demonstrated what their research adds to the literature. Here is a Google Scholar search of the topic for research published since 2020: https://scholar.google.ca/scholar?as_ylo=2020&q=Athletes’+Knowledge+of+Pelvic+Floor+Dysfunction+and+their+2+Knowledge+of+and+Engagement+with+Pelvic+Floor+Muscle+Train-+3+ing:+a+Scoping+Review&hl=en&as_sdt=0,5. There are “About 1,100 results”. The authors must read the most relevant of these and compare and contrast their work. Thank you. We ran this search and identified that many of the papers from 2020 are in our review. We reran the search from 2024 and have added several new perspectives into our discussion.

Additionally, please find supporting citations published since 2020 for those outdated. These include, in the Introduction, citations [6] and [10]. In the Material and Methods, these include [14] (see reference offered above), [15], and [17]. These have all been updated.

Round 2

Reviewer 2 Report

Comments and Suggestions for Authors

Thank you to the authors for the changes they have made to their manuscript. All have improved it. Some remain to be made.

  1. The authors have provided examples of the searches performed in Supporting Information File 1. This information is helpful. However, by not providing the searched keywords for every database searched, the information is insufficient. Please include similar details for all the databases searched.
  2. Figure 1 has no details regarding the 396 studies excluded during screening. The authors must provide details regarding why these studies were excluded.
  3. Table 1 remains needlessly large. Please reduce the font size of the table so that it can be kept within the margins of the manuscript text. Furthermore, please set up Table 1 to correspond with the journal requirements (see the template). Please remove the vertical lines and center the entries.
  4. Please redo the references to correspond with the journal requirements: https://www.mdpi.com/journal/ijerph/instructions.
  5. The authors have said they have included the PRISMA checklist. It is not part of the supplementary files.

Author Response

  1. The authors have provided examples of the searches performed in Supporting Information File 1. This information is helpful. However, by not providing the searched keywords for every database searched, the information is insufficient. Please include similar details for all the databases searched. See supplemental file one for all search terms for all databases.
  2. Figure 1 has no details regarding the 396 studies excluded during screening. The authors must provide details regarding why these studies were excluded. The software used for screening did not save the reasons for exclusion and so we are not able to provide these. Reasons for exclusion for the papers excluded at full text review are provided.
  3. Table 1 remains needlessly large. Please reduce the font size of the table so that it can be kept within the margins of the manuscript text. Furthermore, please set up Table 1 to correspond with the journal requirements (see the template). Please remove the vertical lines and center the entries. This has been completed.
  4. Please redo the references to correspond with the journal requirements: https://www.mdpi.com/journal/ijerph/instructions. This has been completed.
  5. The authors have said they have included the PRISMA checklist. It is not part of the supplementary files. This was provided as a separate file (not for publication). It is now included with the supplementary files.